# Rays as Pixels: Learning A Joint Distribution of Videos and Camera Trajectories

**Wonbong Jang** [1]   **Shikun Liu** [1]   **Soubhik Sanyal** [1]   **Juan Camilo Perez** [1]   **Kam Woh Ng** [1]   **Sanskar Agrawal** [1]
**Juan-Manuel Perez-Rua** [1]   **Yiannis Douratsos** [1]   **Tao Xiang** [1]

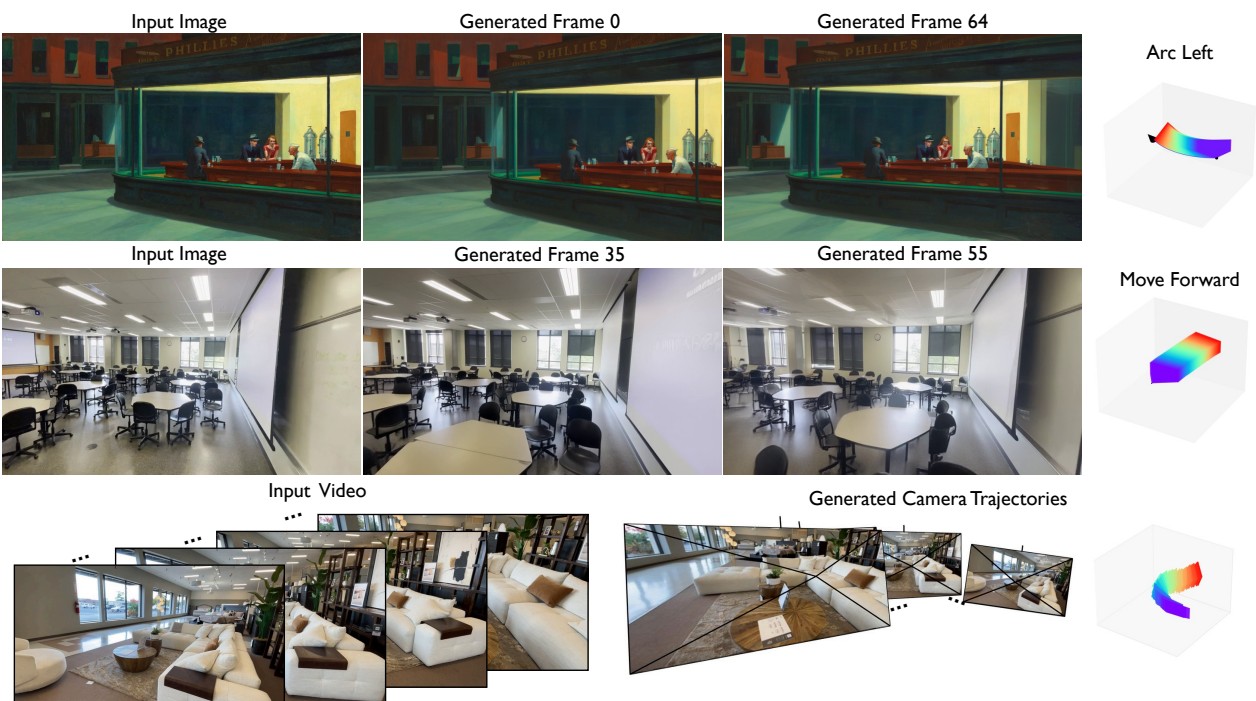

*Figure 1.* **Rays as Pixels: Unifying Video Generation and Camera Pose Estimation.** The first two rows show trajectory-controlled video generation from user-defined camera paths, and the last row shows pose estimation recovering a camera trajectory from a raw input video. By representing camera parameters as dense *raxels* (rays as pixels), our model learns a joint distribution of videos and camera trajectories, enabling both *camera pose prediction and novel view synthesis* within a single model.

## Abstract

Recovering camera parameters from images and rendering scenes from novel viewpoints have been treated as separate tasks in computer vision and graphics. This separation breaks down when image coverage is sparse or poses are ambiguous, since each task depends on what the other produces. We propose Rays as Pixels, a Video Diffusion Model (VDM) that learns a joint distribution over videos and camera trajectories. To our knowledge, this is the first model to predict camera poses and do camera-controlled video generation within a single framework. We represent each camera as *dense ray pixels (raxels)*, a pixel-aligned encoding that lives in the same latent space as video frames, and denoise the two jointly through a *Decoupled Self-Cross Attention* mechanism. A single trained model handles three tasks: predicting camera trajectories from video, generating video from input images along a pre-defined trajectory, and jointly synthesizing video and trajectory from input images. We evaluate on pose estimation and camera-controlled video generation, and demonstrate the model's self-consistency: its predicted poses and the renderings conditioned on those poses agree. Ablations against Plücker embeddings confirm that representing cameras in a shared latent space with

[1]Meta AI, London, United Kingdom. Correspondence to: Wonbong Jang <won.jang1108@gmail.com>.

video is substantially more effective. Project web-page: https://wbjang.github.io/raysaspixels/

## 1. Introduction

Videos implicitly encode 3D geometry through motion and parallax. In 3D computer vision and neural rendering, this manifests as two complementary problems: recovering camera trajectories from images (the inverse process), and rendering images following those trajectories given the underlying geometry (the forward process). Despite this duality, traditional pipelines treat these problems in isolation. Structure-from-Motion (SfM) acts as a prerequisite, demanding dense overlapping views to estimate the camera parameters. Rendering approaches such as Neural Radiance Fields (NeRFs) (Mildenhall et al., 2020) or 3D Gaussian Splatting (3DGS) (Kerbl et al., 2023a) consume predicted camera parameters as ground truth. This separation creates a fragility: when image coverage is sparse or camera motion is ambiguous, the pipeline breaks. This motivates a unified approach: learning the forward and inverse processes jointly within a single generative model.

While optimization-based methods like NeRF and 3DGS excel at reconstructing observed regions, they struggle when input views are scarce. Because they optimize to reconstruct rather than learn to generate, they cannot plausibly synthesize occluded regions, resulting in blurry artifacts or floaters. Recent 3D-aware generative models (Gao et al., 2025; Liu et al., 2026) address this by leveraging data-driven priors to sample plausible completions for unseen regions. In parallel, camera-controlled VDMs (Liang et al., 2024; Bai et al., 2025) generate video along a specified camera trajectory by conditioning on camera parameters alongside dense input frames. However, these approaches share a limitation: they *decouple pose estimation from the generation process.* They treat camera poses as *solved prerequisites*, relying on off-the-shelf estimators, whether classical, such as COLMAP (Schönberger & Frahm, 2016a), or learned, such as DUSt3R (Wang et al., 2024b), to provide conditioning. Consequently, the generative model inherits the fragility of these upstream estimators, which often fail to converge given the very same sparse inputs the generative model is designed to handle.

To overcome this dependency, we introduce a unified VDM that handles both directions of the duality within a single model. Unlike existing 3D-aware generative models, which treat camera parameters as fixed conditioning inputs, our model learns a joint distribution over video frames and camera trajectories. This allows a single trained model to predict camera trajectories from video, generate video along a given trajectory, and jointly synthesize video and trajectory from input images.

The primary challenge is that pretrained video diffusion models operate on dense spatial tensors, while camera parameters are typically low-dimensional global matrices. Prior camera-controlled VDMs handle this mismatch by adding camera-conditioning components, either by injecting camera matrices through learned encoders (Wang et al., 2024c) or by adopting Plücker embeddings (He et al., 2024), and both are fixed conditioning. We instead represent the cameras themselves as RGB images, *rays as pixels (raxels)*, dense maps where every pixel encodes the origin and direction of the corresponding camera ray. We can encode raxels using the same spatio-temporal VAE, and the same diffusion backbone processes them without modification. We further introduce *decoupled self-cross attention* to better couple ray and video latents.

Our model, Rays as Pixels, performs three tasks with a single set of weights:

i) **Camera Pose Estimation:** Recovering camera trajectories from videos, without ground-truth poses or external pose estimators.

ii) **Joint Video and Pose Generation:** Generating video and the corresponding camera trajectory jointly from one or more sparse input images.

iii) **Pose-Conditioned Video Generation:** Generating video that follows a pre-defined camera trajectory, from sparse input images.

We evaluate Rays as Pixels on pose estimation and camera-controlled video generation, validate through a *self-consistency test* that the model's forward and inverse predictions agree, and conduct an ablation study showing that representing a new modality (cameras) in a shared latent space is more effective than direct token-space embedding.

## 2. Related Work

**Reconstructing 3D from Images.** Optimization-based frameworks such as COLMAP (Schönberger & Frahm, 2016b), ORB-SLAM (Mur-Artal & Tardos, 2017), and DROID-SLAM (Teed & Deng, 2021) remain the gold standard for recovering 3D structure and serve as the backbone for creating most real-world 3D datasets. However, these methods rely on dense feature matching, making them brittle under sparse views or limited overlap. Recent feed-forward approaches such as DUSt3R (Wang et al., 2024a), MASt3R (Leroy et al., 2024), VGGT (Wang et al., 2025), Pow3R (Jang et al., 2025) and MapAnything (Keetha et al., 2025) predict 3D pointmaps and camera parameters from sparse views in a single forward pass. While effective at handling sparse inputs, these methods reconstruct rather than learn to generate: they cannot synthesize plausible content for unobserved regions.

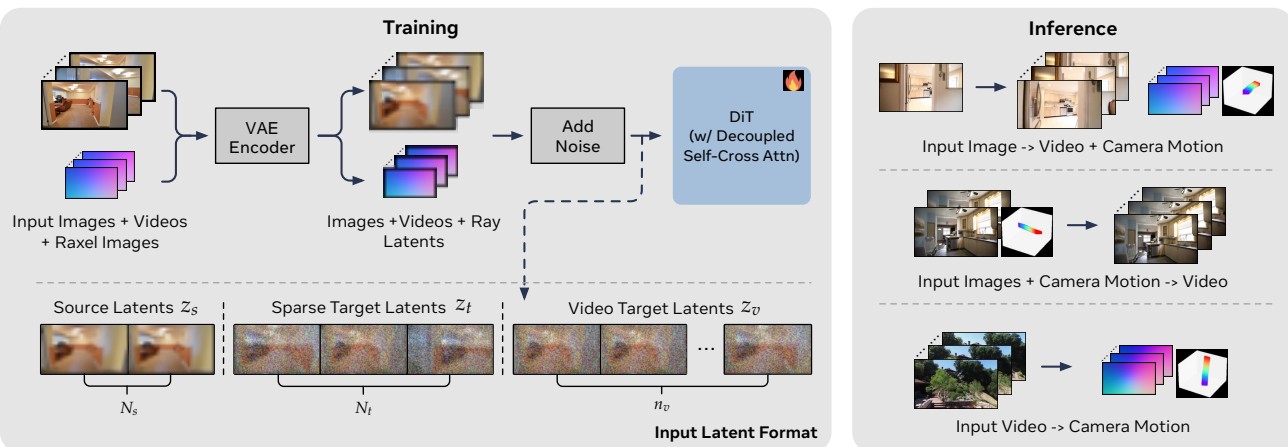

*Figure 2.* **Overview of the Rays as Pixels Framework. Training (left):** We jointly encode video frames and their corresponding *raxel images* into a shared latent space using a frozen spatio-temporal VAE encoder. Video inputs undergo $4\times$ temporal compression, while the temporal dimensions of image inputs remain as they are. The DiT jointly denoises these latents via Decoupled Self-Cross Attention, conditioning on clean source latents while denoising noisy sparse targets. **Inference (right):** A single trained model supports three inference modes: (1) recovering camera trajectories from video; (2) camera-controlled video generation given a target trajectory; and (3) joint generation of a synchronized video and camera trajectory from sparse input images.

**Novel View Synthesis.** NeRF (Mildenhall et al., 2020) and its successors such as 3DGS (Kerbl et al., 2023b) achieve high-quality rendering through per-scene optimization. However, they rely on dense multi-view supervision and lack the generative priors needed to plausibly synthesize occluded regions in sparse-view settings. Feed-forward approaches such as LRM (Hong et al., 2023), pixel-Splat (Charatan et al., 2024), and NViST (Jang & Agapito, 2024) amortize 3D reconstruction across scenes by training transformers on large multi-view datasets, regressing explicit 3D representations from sparse inputs in a single forward pass. Because these methods regress rather than sample, they produce a single deterministic output and cannot capture the multiple plausible completions that occluded regions admit.

**3D-aware Diffusion Models.** Diffusion models have been adapted for 3D tasks through several distinct strategies. Early approaches lifted 2D priors to 3D: DreamFusion (Poole et al., 2022) uses Score Distillation Sampling to optimize NeRFs from text, while Zero-1-to-3 (Liu et al., 2023a) conditions image diffusion models on relative camera poses. More recent multi-view diffusion models (Shi et al., 2024; Liu et al., 2023b; Long et al., 2024; Wu et al., 2024; Gao et al., 2025; Zhou et al., 2025; Liu et al., 2026; Li et al., 2025) generate consistent novel views from sparse inputs by jointly denoising multiple target views. All of these approaches, however, treat camera poses as a fixed conditioning input rather than a generative variable. Unified frameworks for joint pose estimation and view synthesis remain rare; the closest prior work is Matrix3D (Lu et al., 2025), which jointly models RGB, pose, and depth across sparse views using a multi-modal diffusion transformer, then renders final outputs through a 3DGS optimization step. Rays as Pixels differs in two respects. First, we build on

a pretrained video diffusion model and adapt it minimally through our raxel representation, inheriting the temporal priors learned from large-scale video data. Second, we generate video frames directly from the diffusion model, without an intermediate 3D representation or post-processing rendering step.

**Video Diffusion Models.** Scaling diffusion models to the temporal domain has driven rapid progress in video generation. Foundational works such as Make-A-Video (Singer et al., 2022) and Video LDM (Blattmann et al., 2023) established the paradigm of inflating 2D priors to achieve temporal consistency. More recently, large-scale commercial systems like Sora (OpenAI, 2024) and Veo3 (Google Deep-Mind, 2025), alongside open-source models like LTX (Ha-Cohen et al., 2024) and Wan (Wan et al., 2025), have demonstrated that training on massive video data yields high-fidelity, physically plausible results. Beyond pure synthesis, pretrained VDMs have emerged as effective priors with surprisingly broad emergent capabilities (Wiedemer et al., 2025).

**Repurposing Generative Models for Different Tasks.** A growing body of work fine-tunes pretrained generative models for tasks beyond their original objective. Marigold (Ke et al., 2024) adapts Stable Diffusion (Rombach et al., 2021) for monocular depth estimation, and DepthCrafter (Hu et al., 2025) extends this idea to consistent video depth estimation. More broadly, large-scale generative training has been repurposed for interactive world simulation (Bruce et al., 2024). Recovering camera trajectories is different from per-pixel tasks like depth estimation: depth has a one-to-one correspondence with image pixels, while camera trajectories are defined relative to a reference frame and require joint reasoning across the full sequence. Rays as Pixels adapts

a pretrained video diffusion model to *recover camera trajectories from video*, treating the video prior as a basis for both generation and inference while *preserving its video generation capabilities*.

**Camera-Controlled Video Diffusion.** Existing camera-controlled VDMs differ in how they inject camera information. Several methods feed camera matrices through learned adapter modules (He et al., 2024; 2025; Bai et al., 2025), while others adopt dense Plücker embeddings (Bahmani et al., 2025) or epipolar attention (Xu et al., 2024; Kuang et al., 2024). A separate line of work conditions VDMs on explicit 3D representations such as point clouds or 3D Gaussians produced by external estimators (Yu et al., 2025b;a; Liang et al., 2024). Each of these strategies treats cameras as fixed conditioning inputs, often requiring additional adapter layers or external 3D pre-processing. Rays as Pixels instead encodes cameras as raxels, letting the same architecture process them alongside video frames as a generative target rather than a conditioning input.

## 3. Methodology

### 3.1. Preliminaries and Notations

At training time, the input consists of three sets of frames: a target video $V$ of $N$ frames, $N_s$ source images, and $N_t$ sparse target images, with the total $N + N_s + N_t$ held fixed across training examples. The source images provide clean conditioning, the sparse target images keep the input length fixed regardless of how many source images are provided, and the target video provides dense temporal supervision. Each frame $I \in \mathbb{R}^{H \times W \times 3}$ comes with an intrinsic matrix $K \in \mathbb{R}^{3 \times 3}$ and an extrinsic (camera-to-world) matrix $P \in \mathbb{R}^{4 \times 4}$. We use subscripts $s$ and $t$ to denote source and sparse target frames respectively.

We encode the visual inputs into a lower-dimensional latent space using a pretrained spatio-temporal VAE (VAE) encoder $\mathcal{E}$. The target video $V$ is encoded into video latents $z_v \in \mathbb{R}^{n_v \times h \times w \times c}$, with $4\times$ temporal compression so that $N = 4(n_v - 1) + 1$. The source and sparse target images are encoded by the same VAE without temporal compression, yielding source latents $z_s$ of length $N_s$ and sparse target latents $z_t$ of length $N_t$. The source latents $z_s$ remain clean throughout training and serve as conditioning, while the sparse target latents $z_t$ are noised and supervised together with the video latents $z_v$.

### 3.2. Representing Rays as Pixels

To make camera parameters compatible with a pretrained video diffusion model, we represent each camera as a **raxel image**: a dense per-pixel map where every pixel encodes the ray origin and direction associated with that pixel. Because raxel images share the same 3-channel spatial structure as

RGB video frames, we can encode them using the same pretrained VAE encoder $\mathcal{E}$, placing raxel images and their corresponding video frames in a shared latent space. Table 1 compares raxels with alternative camera encodings: low-dimensional representations such as camera matrices lack spatial structure, Plücker embeddings (Zhang et al., 2024) and raymaps (Lin et al., 2025; Zhao et al., 2025) are dense but their 6 channels are incompatible with 3-channel pre-trained VAEs, and pointmaps proposed by DUSt3R (Wang et al., 2024a) require per-pixel depth that is not available at inference time.

*Table 1.* **Comparison of Camera Representations.** We define the ray direction $\mathbf{d} = X/\|X\|$, where $X$ denotes the unprojected pixel coordinates obtained from the intrinsic matrix $K$. $R$ and $T$ denote rotation and translation, $D$ denotes depth, and $\times$ denotes the cross product. Raxels are the only representation that matches both the spatial structure and the 3-channel input format of a pretrained video diffusion VAE.

| Representation | Formulation | Dimension | Limitation |
|---|---|---|---|
| Camera Matrices | $K, R, T$ | $(4, 4)$ | spatially unaligned |
| Plücker Embedding | $[R\mathbf{d}, R\mathbf{d} \times T]$ | $(H, W, 6)$ | input channel mismatch |
| Raymap | $[T, R\mathbf{d}]$ | $(H, W, 6)$ | input channel mismatch |
| Pointmap | $RX \cdot D + T$ | $(H, W, 3)$ | depth required |
| **Raxels (Ours)** | $R\mathbf{d} + T$ | $(H, W, 3)$ | — |

**Coordinate Canonicalization.** To ensure the model learns generalizable relative poses rather than memorizing absolute global coordinates, we canonicalize the scene to a randomly selected reference frame, indexed $s$. We then transform the extrinsic matrices of all video, source, and target frames into its coordinate system:

$$P_{\text{rel}}^{(j)} = P_s^{-1} P_j. \tag{1}$$

This places the reference camera at the origin with identity rotation ($P_{\text{rel}}^{(s)} = \mathbf{I}$), decoupling the camera parameters from the arbitrary global coordinate frame of the original dataset.

**Raxel Image Construction.** For every pixel coordinate $\mathbf{u} = [u, v]^\intercal$ in a frame $I_j$, we un-project it to the camera coordinate system using the intrinsic matrix $K_j$, normalize the resulting direction, and then transform it to the canonical world space using the relative extrinsic $P_{\text{rel}}^{(j)}$, composed of rotation $R_{\text{rel}}^{(j)}$ and translation $T_{\text{rel}}^{(j)}$. This yields the world-space ray direction $\mathbf{d} \in \mathbb{R}^3$ and origin $\mathbf{o} \in \mathbb{R}^3$:

$$\mathbf{d} = R_{\text{rel}}^{(j)} \frac{K_j^{-1} \tilde{\mathbf{u}}}{\left\| K_j^{-1} \tilde{\mathbf{u}} \right\|_2}, \quad \mathbf{o} = T_{\text{rel}}^{(j)}, \tag{2}$$

where $\tilde{\mathbf{u}}$ is the homogeneous pixel coordinate. We define the *raxel* at coordinate $\mathbf{u}$ as the vector sum $\mathbf{d} + \mathbf{o}$, a compact 3-channel encoding that combines the ray's direction and origin into a single RGB-compatible representation. These per-pixel vectors form the dense raxel image $R_j \in \mathbb{R}^{H_r \times W_r \times 3}$, which we compute on a coarser pixel grid ($H_r = H/2, W_r = W/2$) to reduce token overhead.

| Input Image | Kaleido | Ours | Ground Truth |
|---|---|---|---|

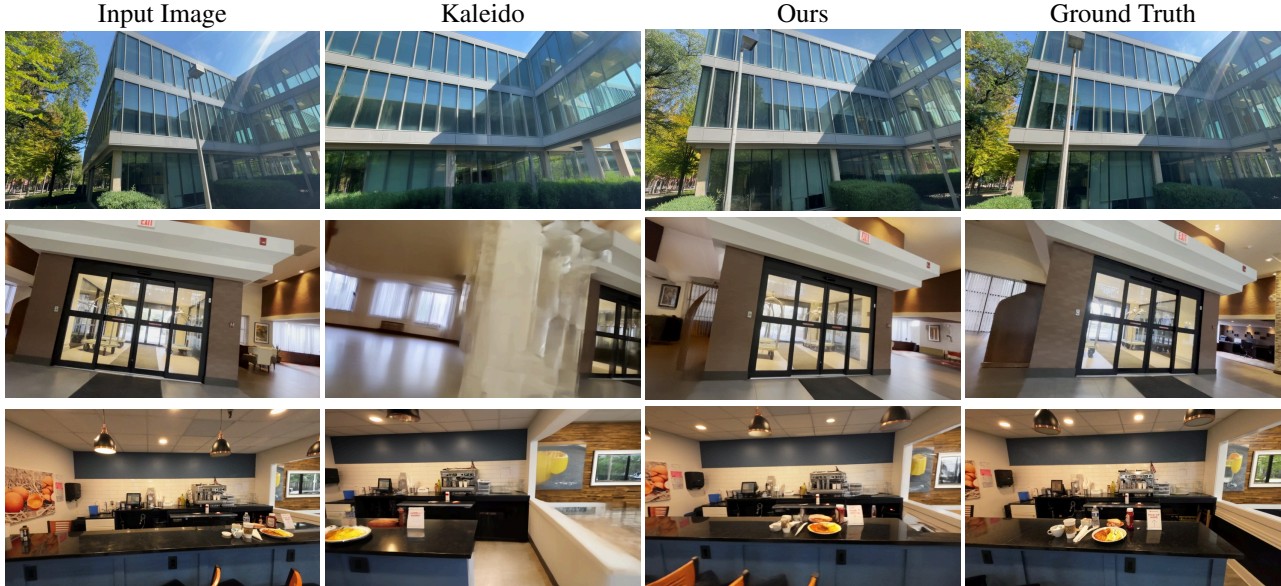

*Figure 3.* **Qualitative Results on DL3DV.** We visualize the NVS performance of our model against the state-of-the-art baseline Kaleido (Liu et al., 2026) on DL3DV. Given a single reference image (first column), our model synthesizes target views (third column) that exhibit superior structural fidelity and lighting consistency, closely matching the ground truth (fourth column). Note that ours predicts the camera parameters by itself.

We then encode each raxel image using the shared VAE encoder $\mathcal{E}$, producing latents $r_v, r_s, r_t$ that spatially align with the video latents $z_v, z_s, z_t$. Because raxel latents and video latents share the same spatial grid, we apply the same Rotary Positional Embedding (RoPE) (Su et al., 2021) to both, with an additional learnable modality embedding to distinguish video latents from raxel latents.

### 3.3. Joint Denoising via Flow Matching

Given the visual latents $z \in \{z_v, z_s, z_t\}$ and their corresponding ray latents $r \in \{r_v, r_s, r_t\}$, we model the joint distribution $p(z, r)$. Previous methods typically learn the conditional distributions $p(z|r)$ for view synthesis or $p(r|z)$ for pose estimation; we instead learn the joint density, which lets a single model recover camera trajectories from video, generate video from camera trajectories, and jointly generate both from input images.

**Flow Matching Formulation.** We adopt Flow Matching (Lipman et al., 2023) as our generative framework: it integrates cleanly with our pretrained backbone and provides straight-line sampling paths that require fewer steps than DDPM-style diffusion at inference. Let $x = [z, r]$ denote the concatenation of visual and ray latents. We define a time-dependent probability density path $p_t(x)$ that transforms a Gaussian prior $p_0(x) = \mathcal{N}(\mathbf{0}, \mathbf{I})$ at $t = 0$ into the data distribution $p_1(x) \approx p_{\text{data}}(z, r)$ at $t = 1$. The conditional probability path is a linear interpolation:

$$x_t = (1-t)x_0 + tx_1, \quad t \in [0, 1], \quad x_0 \sim p_0. \quad (3)$$

This corresponds to a constant conditional velocity field $u_t(x \mid x_1) = x_1 - x_0$, guiding the flow from noise to data along a straight line.

**Unified Tokenization.** We concatenate visual latents $z$ and ray latents $r$ along the sequence dimension. Because the two modalities occupy distinct token positions, we can apply asymmetric inference schedules: for example, fixing $r$ at $t = 1$ for camera-conditioned video generation while denoising $z$. We find that $r$ converges much faster than $z$ during sampling, and a few denoising steps on $r$ achieves comparable results to multi-step inference on self-consistency test (Table 3).

**Training Objective.** We parameterize the velocity field with a neural network $v_\theta(x_t, t)$ that predicts the target velocity $u_t = x_1 - x_0$. The velocity vector has both a magnitude and a direction; while the MSE loss penalizes both jointly, we add a cosine similarity term that isolates the direction component:

$$\mathcal{L}(\theta) = \mathbb{E}_{t, x_0, x_1}\left[\|v_\theta - u_t\|^2 + \lambda\left(1 - \frac{v_\theta^\top u_t}{\|v_\theta\|\|u_t\|}\right)\right], \quad (4)$$

where $\lambda$ weights the cosine term (set to $0.5$ in our experiments). We find that the cosine term improves self-consistency in our ablations (Table 3).

### 3.4. Decoupled Self-Cross Attention

While ray latents $r$ and video latents $z$ inhabit the same VAE latent space, they have different structural characteristics. Visual latents $z$ encode dense, high-frequency texture,

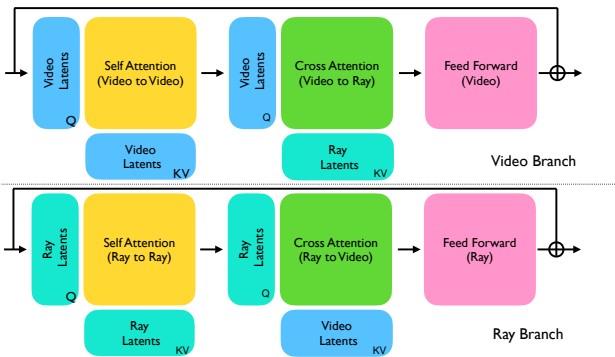

*Figure 4.* **Decoupled Self-Cross Attention.** We replace standard self-attention in the video diffusion backbone with a *Decoupled Self-Cross Attention* block that processes video and ray latents in parallel branches. Within each branch, *intra-modal self-attention* operates on tokens of the same modality, followed by *inter-modal cross-attention* where queries from one modality attend to keys and values from the other. This separation encourages stable training and allows the video and ray latents to attend to each other.

while ray latents $r$ encode smooth camera ray information anchored to a canonical reference frame. The two modalities also differ in their temporal profiles: visual content can change rapidly between frames due to camera motion, occlusion, and lighting, while the underlying camera trajectory evolves smoothly relative to the reference frame $s$. We find that applying a single global self-attention across the concatenated sequence under-utilizes this asymmetry: the model tends to fit each modality somewhat independently rather than learning rich cross-modal dependencies.

To address this, we replace the standard self-attention with a *Decoupled Self-Cross Attention*. Each transformer block applies attention in two stages: **i) Self-Attention (Intra-Modal):** we apply self-attention separately to $z$ and $r$. This enforces consistency within each modality, ensuring temporal smoothness in video frames and trajectory coherence in ray paths, without interference between modalities. **ii) Symmetric Cross-Attention (Inter-Modal):** cross-attention layers then exchange information between modalities, with visual tokens attending to ray tokens ($z \leftarrow r$) and ray tokens attending to visual tokens ($r \leftarrow z$). This guides the video generation to follow the predicted camera rays while allowing the ray latents to refine their trajectory based on visual context.

One subtlety: without positional encoding, cross-attention treats keys and values as an unordered set. Because raxel latents and video latents are spatially aligned at every position, we need to preserve this alignment in the cross-attention. We apply RoPE to both queries and keys, ensuring that each visual token attends to its spatially corresponding raxel token (and vice versa).

**Probabilistic Interpretation.** This decomposition has a clean probabilistic motivation. By the chain rule of probability, the joint log-likelihood $\log p(z, r)$ factorizes in two

equivalent ways:

$$\log p(z, r) = \underbrace{\log p(r)}_{\text{Self-Attn}(r)} + \underbrace{\log p(z|r)}_{\text{Cross-Attn}(z \leftarrow r)} \equiv \underbrace{\log p(z)}_{\text{Self-Attn}(z)} + \underbrace{\log p(r|z)}_{\text{Cross-Attn}(r \leftarrow z)}.$$
(5)

By decoupling these operations, we encourage the model to learn the marginal priors (via self-attention) and the conditional dependencies (via cross-attention) separately. This is consistent with a related observation from *Seeing without Pixels* (Xue et al., 2025), which shows that camera information can be incorporated into video models as a lightweight modality on top of the existing visual backbone.

### 3.5. From Raxels to Camera Parameters

**Pose Recovery via Procrustes Alignment.** We decode the ray latents $r_v, r_s, r_t$ using the pretrained VAE decoder, obtaining reconstructed raxel images $\hat{R}_v, \hat{R}_s, \hat{R}_t$. Recall that raxel images are expressed in a canonicalized coordinate system anchored at the source frame $I_s$ (Section 3.2). The reference raxel image $\hat{R}_s$ represents the bundle of camera rays at the identity pose ($P_{\text{rel}}^{(s)} = \mathbf{I}$), and any other frame $k$ contains the same bundle transformed by the relative pose $P_{\text{rel}}^{(k)}$. We recover this pose by aligning the predicted bundle $\hat{R}_k$ to the reference bundle $\hat{R}_s$ as a rigid registration problem:

$$\hat{P}_{\text{rel}}^{(k)} = \underset{P \in SE(3)}{\arg\min} \sum_{i,j} \left\| \hat{R}_k^{(i,j)} - P\hat{R}_s^{(i,j)} \right\|^2,$$
(6)

where $(i, j)$ ranges over the spatial grid of the raxel image. This admits a closed-form solution via Orthogonal Procrustes (Luo & Hancock, 1999; Brégier, 2021).

**Focal Length Recovery via Median-of-Ratios.** Once $\hat{P}_{\text{rel}}^{(k)}$ is recovered, we transform $\hat{R}_k$ back into its own camera coordinate system by applying the inverse pose, yielding per-pixel local ray vectors $\mathbf{d}_{\text{local}}^{(i,j)} = (\hat{P}_{\text{rel}}^{(k)})^{-1}\hat{R}_k^{(i,j)}$. Under a pinhole camera model, each local ray $(x, y, z)$ relates to its centered pixel coordinate $(u, v)$ by $u/f_x = x/z$ and $v/f_y = y/z$. To recover the focal lengths robustly against decoding noise, we use the Median-of-Ratios estimator:

$$\hat{f}_x = \underset{i,j}{\text{median}} \left( \frac{u_{i,j} \cdot \hat{z}_{i,j}}{\hat{x}_{i,j}} \right), \quad \hat{f}_y = \underset{i,j}{\text{median}} \left( \frac{v_{i,j} \cdot \hat{z}_{i,j}}{\hat{y}_{i,j}} \right) \tag{7}$$

where $(u_{i,j}, v_{i,j})$ are pixel coordinates centered at the principal point (assumed to be the image center).

## 4. Experiments

**Architecture Design.** We build on the Wan 2.1 14B Text-to-Video model (Wan et al., 2025). We choose the T2V

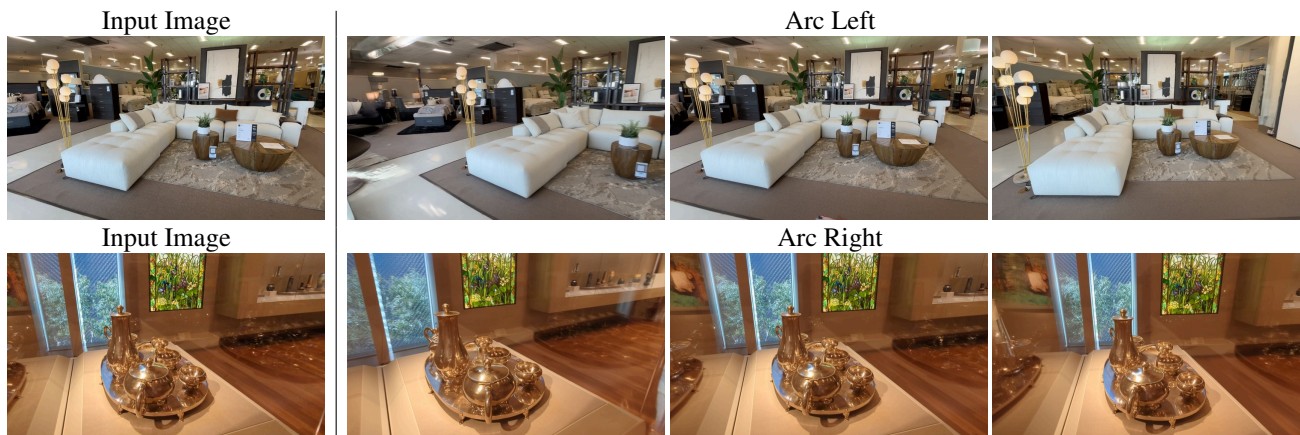

*Figure 5.* **Qualitative Results on DL3DV-140 following the predefined camera trajectory.** We visualize pose-conditioned video generation given a single input image from DL3DV-140 (Ling et al., 2024) and a specific camera path. These scenes present distinct challenges: the top example features a cluttered layout with an off-center subject, while the bottom example contains highly reflective metallic surfaces. **Top:** The model renders the scene from "Arc Left" camera movement while maintaining the geometric consistency of sofa and many objects within a frame. **Bottom:** The model is capable of synthesizing the scene from "Arc Right" trajectory, notably our model can synthesize view-dependent reflections on the metallic surfaces while following the camera path.

checkpoint rather than the Image-to-Video (I2V) variant because I2V is post-trained from T2V with a fixed frame ordering that conflicts with our flexible source-frame setup. We replace the standard self-attention in each transformer block with our decoupled self-cross attention (Section 3.4), and add a dedicated ray branch consisting of independent layer normalizations, feed-forward networks, and linear embedding layers for the ray latents. The ray branch is initialized from the corresponding pretrained video layers and adds 6B parameters, for a total of 20B. We fine-tune all parameters during training.

**Training Datasets and Strategies.** We train on two real-world datasets: **RealEstate10K** (Re10K) (Zhou et al., 2018) and **DL3DV** (Ling et al., 2024). The camera parameters for these datasets, obtained from ORB-SLAM and COLMAP respectively, are reconstructed at arbitrary per-scene scales. We align all scenes to a common metric scale using (Hu et al., 2024; Keetha et al., 2025).

We resize and center-crop both datasets to $480 \times 832$ while preserving the original aspect ratio. To encourage scene–trajectory disentanglement, we apply time-reversal augmentation: each scene's trajectory is augmented with its reverse, giving the model two distinct trajectories per scene during training.

**Qualitative Results.** Figures 1, 3, 5, and 6 together demonstrate the visual quality and versatility of our model. Figure 1 shows the model generating consistent frames from artwork and our test set following pre-defined camera trajectories, as well as predicting camera trajectories within the same model. Figure 3 compares Rays as Pixels against Kaleido (Liu et al., 2026) on DL3DV-140 test set, where our model produces better results. Figure 5 shows our model following predefined camera trajectories on complex scenes

from DL3DV-140 test set: the top row demonstrates consistency in cluttered spatial arrangements, and the bottom row shows view-dependent effects such as specular reflections on metallic surfaces. Figure 6 visualizes the cycle self-consistency test under different ablation conditions, where replacing raxels with Plücker embeddings causes the most visible degradation. Unlike I2V models, our approach is flexible in the temporal position of the input image, which can serve as the first, middle, or last frame of the generated sequence.

## 4.1. Self-Consistency and Ablation Study

Our model learns a joint distribution over video latents $z$ and camera trajectories $r$, which enables a test that no conditional-only model can pass: *cycle self-consistency*. A model that learns only $p(z \mid r)$ or only $p(r \mid z)$ cannot round-trip through both directions, because its forward and inverse predictions are not constrained to agree. We use cycle consistency both as a direct evaluation of our joint distribution learning and as the metric for our ablation study.

**Cycle Setup.** We sample a ground-truth pair $(z, r)$ and select three source images from $z$ as conditioning. We first predict the camera trajectory by sampling $r' \sim p(r \mid z)$, then re-generate the video conditioned on the predicted trajectory and the three source images, $z' \sim p(z \mid r', I_s)$. A self-consistent model satisfies two properties: (i) $r'$ is close to the ground-truth trajectory $r$, and (ii) $z'$ is close in distribution to $z$. We measure (i) with rotation error $R_{\text{err}}$ and translation error $T_{\text{err}}$ computed against ground truth, and (ii) with FID and FVD on the re-generated set. We evaluate on the DL3DV-140 test set, sampled at 12 FPS directly from the raw videos rather than the standard DL3DV-140 COLMAP split, which uses non-uniform subsampling.

| Plücker | No DSCA | No Cosine | Ours | Ground Truth |
|---|---|---|---|---|

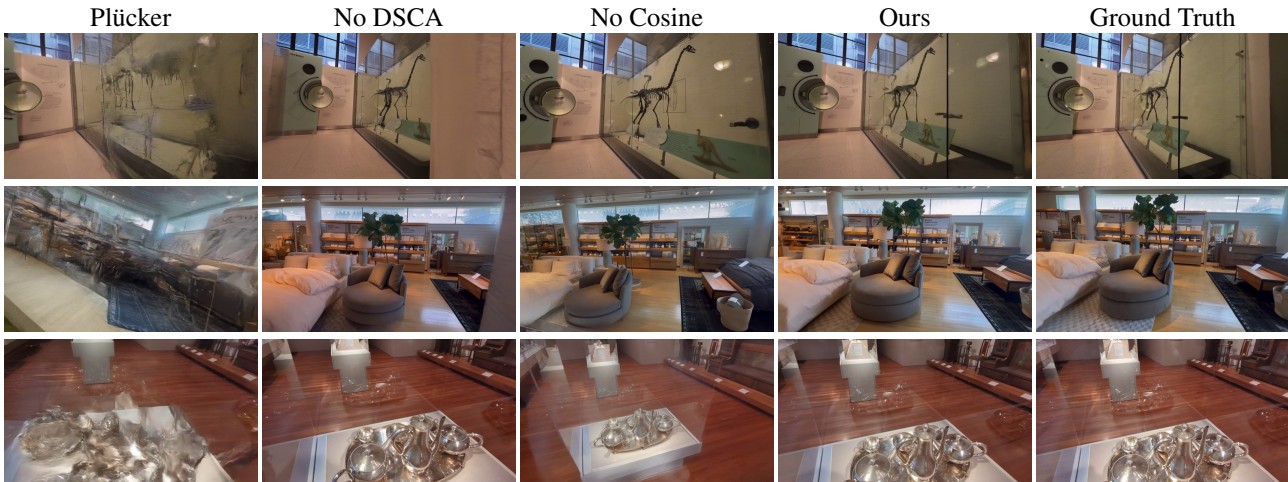

*Figure 6.* **Self-Consistency and Ablation Study.** Re-generated frames from the cycle self-consistency test on three scenes from DL3DV. From left to right: with Plücker embeddings replacing raxels, without DSCA, without cosine similarity loss, our full model, and ground truth. Replacing raxels with Plücker embeddings causes the most severe degradation, consistent with the quantitative results in Table 2.

**Ablation Study.** We ablate three design choices using cycle self-consistency as the metric (Table 2): the decoupled self-cross attention (DSCA), the cosine similarity loss, and the choice of ray representation. For the ray-representation ablation, we replace raxels with Plücker embeddings: since Plücker embeddings have 6 channels and cannot be encoded by the pretrained VAE, we instead embed them directly via an MLP and concatenate the result with the VAE-encoded video latents. Since VAE temporally compresses by 4, we linearly interpolate camera embeddings from Plückers to match temporal dimensions in pixel spaces.

**Results.** Table 2 shows that the Plücker-embedding variant dramatically underperforms raxels across every metric: FID (Heusel et al., 2017), FVD (Unterthiner et al., 2018; Skorokhodov et al., 2022), rotation error, and translation error. These gaps hold despite the architecture being otherwise identical except for the ray encoding. The raxel representation, by encoding both ray origin and direction in a form compatible with the pretrained VAE, gives the model a substantially better starting point than Plücker's 6-channel input-level embedding.

Removing the cosine similarity loss or DSCA also hurts self-consistency, though less dramatically across every metric. The metric-scale training data provides a well-conditioned signal for cross-modal learning, making DSCA a useful refinement rather than a strict requirement. Still, the raxel representation and metric-scale training are the primary load-bearing design choices.

### 4.2. Camera Pose Estimation

**Experimental Setup.** We evaluate pose estimation on three benchmarks: 975 quality-filtered clips from Re10K (Zhou et al., 2018), all 299 test clips from DL3DV-140 (Ling et al., 2024), and all 211 clips from Tanks and

*Table 2.* **Self-Consistency and Ablation Study.** We evaluate cycle consistency on DL3DV (sampled at 12 FPS from raw videos) with three source images. Each row shows the effect of removing or replacing one design choice. Replacing raxels with Plücker embeddings degrades all four metrics by large margins, while removing DSCA or the cosine similarity loss produces smaller but consistent drops. Lower is better for all metrics.

| Method | FID ↓ | FVD ↓ | $R_{\text{err}}$ ↓ | $T_{\text{err}}$ ↓ |
|---|---|---|---|---|
| **Ours** | **7.33** | **68.17** | **0.020** | **0.018** |
| w/o DSCA | 8.69 | 77.08 | 0.048 | 0.052 |
| w/o Cosine Sim. Loss | 9.48 | 97.84 | 0.058 | 0.094 |
| Plücker Embedding | 21.97 | 333.56 | 0.241 | 0.430 |

Temples (T&T) (Knapitsch et al., 2017). These benchmarks cover two distinct regimes: Re10K and T&T contain temporally continuous video clips where consecutive frames are close viewpoints, while DL3DV-140 follows the standard COLMAP subsampling convention, producing wide-baseline frames with large pose differences. Our model is trained on relatively continuous video sequences, so DL3DV-140 probes generalization to a sampling regime different from training. We compare against VGGT (Wang et al., 2025), a feed-forward pose estimator trained on wide-baseline multi-view data, evaluated on the same frames for fair comparison.

We report mean Relative Rotation Accuracy (mRRA@30), computed over all frames in each clip. Note that we do not report Relative Translation Accuracy: both the ground-truth camera parameters and VGGT are scale-ambiguous, so angular translation distance is very sensitive to small changes near the reference frame $I_s$.

**Results.** Table 3 shows two main findings. First, across all three benchmarks, our model reaches its best rotation accuracy at 2 diffusion steps, with further steps producing slightly worse results. The pattern is consistent: 1

*Table 3.* **Camera Pose Estimation.** Mean Relative Rotation Accuracy (mRRA@30, $\times 100$) across diffusion step counts on three benchmarks, with VGGT (Wang et al., 2025) as a reference. Our model reaches its best accuracy at 2 diffusion steps across all benchmarks. Higher is better.

| Dataset | 1 step | 2 steps | 5 steps | 20 steps | VGGT |
|---|---|---|---|---|---|
| Re10K | 95.39 | **95.91** | 95.08 | 94.78 | **98.07** |
| DL3DV-140 | 82.78 | **88.37** | 87.30 | 85.90 | **91.86** |
| T&T | 92.34 | **93.51** | 93.43 | 93.01 | **97.70** |

step underperforms, 2 steps is the peak, and 5 or 20 steps degrade gradually. This is consistent with the observation in Section 3.3 that ray latents converge much faster than video latents under joint flow matching. Second, VGGT (Wang et al., 2025) achieves higher rotation accuracy across all benchmarks. This is expected: VGGT predicts 3D pointmaps, which inherently encode depth and camera poses, so pose parameters can be derived directly from the pointmap output. Our model predicts videos, which do not contain explicit geometric information; camera parameters are learned as a separate modality (raxels) jointly with video generation but without explicit 3D representations.

### 4.3. Camera-Controlled Video Generation

**Experimental Setup.** We evaluate visual quality FID (Heusel et al., 2017) and temporal coherence FVD (Unterthiner et al., 2018; Skorokhodov et al., 2022) on Re10K, DL3DV-140, and Tanks and Temples, following the protocols of Wonderland (Liang et al., 2024) and Kaleido (Liu et al., 2026), using VGGT (Wang et al., 2025) to validate the trajectory adherence. Since the ground-truth trajectories are scale-ambiguous, we use the variant of our model trained with per-scene normalized camera trajectories for this evaluation. We compare against MotionCtrl (Wang et al., 2023) and VD3D (Bahmani et al., 2025), which condition video diffusion models on camera matrices and Plücker embeddings; ViewCrafter (Yu et al., 2025b) and Wonderland (Liang et al., 2024), which integrate explicit 3D representations; and Kaleido (Liu et al., 2026), an image-based generative novel view synthesis model with camera positional embedding.

**Results.** Table 4 shows that our model achieves the best FID and FVD on all three benchmarks without using explicit 3D representations or camera-specific positional embeddings.

### 4.4. Limitations

First, our training data consists of static scenes with smooth camera trajectories, so the model may not generalize well to rapid camera motion or scenes containing dynamic objects. Second, the $4\times$ temporal compression of the VAE compresses consecutive frames into shared latent positions, which leads to the loss on temporal dimensions.

*Table 4.* **Quantitative Comparison of Camera-Controlled Video Generation Evaluations.** We evaluate visual quality (FID $\downarrow$, FVD $\downarrow$) and trajectory adherence ($R_{err} \downarrow$, $T_{err} \downarrow$) across three benchmarks, and achieve the best performance on visual qualities (FID, FVD) with temporal coherence on generated videos.

| Method | Metrics | | | |
|---|---|---|---|---|
| *Dataset* | FID $\downarrow$ | FVD $\downarrow$ | $R_{err} \downarrow$ | $T_{err} \downarrow$ |
| *RealEstate10K* | | | | |
| MotionCtrl | 22.58 | 229.34 | 0.231 | 0.794 |
| VD3D | 21.40 | 187.55 | 0.053 | 0.126 |
| ViewCrafter | 20.89 | 203.71 | 0.054 | 0.152 |
| Wonderland | 16.16 | 153.48 | **0.046** | **0.093** |
| Kaleido | 18.04 | 103.03 | 0.049 | 0.181 |
| Ours | **15.76** | **98.72** | 0.056 | 0.115 |
| *DL3DV-140* | | | | |
| MotionCtrl | 25.58 | 248.77 | 0.467 | 1.114 |
| VD3D | 22.70 | 232.97 | 0.094 | 0.237 |
| ViewCrafter | 20.55 | 210.62 | 0.092 | 0.243 |
| Wonderland | 17.74 | 169.34 | 0.061 | 0.130 |
| Kaleido | 41.18 | 458.60 | **0.011** | **0.026** |
| Ours | **9.73** | **102.52** | 0.098 | 0.192 |
| *Tanks and Temples* | | | | |
| MotionCtrl | 30.17 | 289.62 | 0.834 | 1.501 |
| VD3D | 24.33 | 244.18 | 0.117 | 0.292 |
| ViewCrafter | 22.41 | 230.56 | 0.125 | 0.306 |
| Wonderland | 19.46 | 189.32 | 0.094 | 0.172 |
| Kaleido | 14.84 | 245.09 | **0.016** | **0.086** |
| Ours | **13.02** | **187.03** | 0.105 | 0.192 |

## 5. Conclusion

Camera pose estimation and camera-controlled video generation have traditionally been treated as separate tasks, with pose estimators and video generators trained and evaluated independently. We presented Rays as Pixels, a unified framework that learns the joint distribution of video frames and camera trajectories in a single model. By encoding camera parameters as dense raxel images compatible with a pretrained video VAE, and by introducing decoupled self-cross attention to couple the two modalities, we enable joint denoising of video and ray latents. Our experiments show visual quality on camera-controlled video generation benchmarks, competitive pose estimation accuracy. We also demonstrate strong cycle self-consistency: the model can predict a camera trajectory from a video and re-generate the video from its own predicted trajectory with minimal degradation, a property that single conditional models cannot achieve.

More broadly, we view the raxel representation as an instance of a general pattern: a non-visual modality (such as camera, segmentation) re-encoded as an image-compatible tensor that shares a pretrained visual backbone's latent space. More speculatively, the joint distribution of visual observations and camera motion that our model captures is one of the primitives relevant to embodied perception, where agents must reason about both what they see and how they are moving through the world.

## Impact Statement

This work models the joint distribution of video sequences and camera trajectories, with applications to camera-controlled video generation, pose estimation, and view synthesis. Like other advances in generative video models, our method could be misused to create misleading or fabricated content, although it does not introduce risks beyond those already present in existing video generation systems. We support continued work on detection tools, provenance tracking, and safety protocols to address these concerns as generative models become more capable.

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

## A. Normalizing the Dataset

The spatio-temporal VAE we build on expects its inputs to lie in the range $[-1, 1]$, the same range used for the RGB video frames. Because we encode camera rays as raxel images and pass them through this shared encoder without modification, the raxels must be brought into the same range before encoding. Unlike pixel intensities, however, ray origins and directions have no intrinsic bound: their magnitudes depend on the scene scale and on how the camera trajectory is parameterized. We therefore normalize the raxels explicitly, and the appropriate strategy differs depending on whether the dataset is defined in a normalized scale or in metric scale.

For the normalized-scale dataset, scene scale is arbitrary and not directly comparable across scenes, so we normalize on a per-scene basis: for each scene we rescale its raxel images so that their values fall within $[-1, 1]$. This keeps each scene's geometry well conditioned for the VAE regardless of the absolute units in which that scene was captured.

For the metric-scale dataset, the absolute scale carries meaning and must remain consistent across scenes—rescaling each scene independently would discard exactly the metric information we wish to preserve. We therefore apply a single global normalization computed over the entire dataset, choosing one scaling factor such that all raxel values across all scenes lie within $[-1, 1]$. This retains the relative scale between scenes while still satisfying the VAE's input requirement.

## B. Decoupled Self-Cross Attention

Our model applies the same RoPE to both video and ray latents, so two tokens from different modalities at the same position share an identical positional embedding. This can lead to suboptimal results, such as artifacts from one modality leaking into the other. We found that this failure is less prominent on the metric-scale dataset than on the normalized-scale dataset. With the metric-scale dataset, raxel images occupy different value ranges depending upon each scene, whereas on the normalized-scale dataset every scene is rescaled to $[-1, 1]$; the wider spread of raxel values makes it easier for the network to distinguish the two modalities even when they share the same positional embedding. Decoupled Self-Cross Attention (DSCA) addresses this directly by separating the attention pathways, so that each modality attends within itself while still exchanging information across modalities in a controlled way.

DSCA can be seen as a lightweight, single-backbone alternative to dual-branch designs for multimodal joint modeling. DUSt3R (Wang et al., 2024b), for instance, processes its two views with a shared ViT encoder but then routes them through two separate decoder branches that exchange information via cross-attention, while LTX-2 (HaCohen et al., 2026) couples a dedicated 14B video stream and a 5B audio stream through bidirectional cross-attention. In contrast, because all of our modalities are represented in RGB space, we do not introduce a second branch or stream: we keep a single shared network and pretrained backbone, and decouple only the attention. This preserves the parameter count and the pretrained weights while still allowing the model to treat video and ray latents distinctly.

## C. Joint Training with Videos without Camera Annotations

Rays as Pixels can be trained jointly on dynamic videos that lack camera annotations. Because acquiring real-world video datasets with accurate camera parameters does not scale, pose estimation is brittle on dynamic scenes and exhaustive annotation is expensive, restricting training to fully posed data would severely limit the available data. We therefore mix two sources of supervision: static videos with their raxels, trained with the full objective including the raxel loss, and dynamic videos without camera annotations, trained with the raxel loss disabled. The posed static data teaches the model the joint video–camera distribution, while the unposed dynamic data broadens its coverage of real-world motion and content.

