# OpenReview forum: "Rays as Pixels: Learning A Joint Distribution of Video and Camera Trajectories"
_ICML.cc/2026/Conference — ICML 2026 regular_

### Official Review · Reviewer_zWsM · 2026-03-06

**Soundness:** 3
**Presentation:** 3
**Significance:** 3
**Originality:** 3
**Overall Recommendation:** 4
**Confidence:** 4

**Summary:**

This work marry NVS and camera pose prediction tasks together through unified representation and joint denoising. The proposed model is novel in the sense of ability to do two tasks at the same time. The weakness is architecture design and lack of motivation.

**Compliance With Llm Reviewing Policy:**

Affirmed.

**Final Justification:**

My concerns are largely solved. In the rebuttal authors strengthened the motivation of this paper, which is my biggest concern.

One thing I would suggest is to weaken the contribution of Raxel since 3 channels representation is better than plucker only under specific constraint, which might not transfer to other settings.

I would like to raise my score to 4 therefore.

**Key Questions For Authors:**

listed above.

**Limitations:**

listed above.

**Strengths And Weaknesses:**

strengths:
1. novelty: it seems the first model designed to handle two tasks at the same time, through unified representation.

weakness:
1. motivation: it makes sense to merge if one can merry two works together and achieve descent result for both. But from the table 2. it seems fall much behind the VGGT on more complex datasets, i.e. DL3DV or TnT. Then could you convince me why we want to "bridge the gap between perceiving camera trajectories and rendering novel views within a single generative framework" and use your model on camera pose estimation task instead of VGGT?
2. sus baseline: in table 3. DL3DV-140 it seems weird that all baselines have very high FID & FVD while have similar or better Rerr and Terr than yours. Could you explain why?
3. lacking ablation study: could you justify following questions with exp results?
a. why your raxel is better than plucker? I dont see anything special with Raxel representation. If it is a design that works better empirically then you should show the evidence through exp results.
b. why using naive attention causes modality collapse? why do you use sequence dimension concatenation for you can cat along feature dimension, use some feature mixer then those two uncommon designs could gone? also, If it is a design that works better empirically then you should show the evidence through exp results.

---

> ### Author Rebuttal · Authors · 2026-03-30
>
> We thank the reviewer for recognizing the novelty of our unified framework. We address each concern below.
>
> ### W1: Motivation: why use this model for pose estimation instead of VGGT?
>
> Our claim is not that our model should replace dedicated methods like VGGT. Rather, the key contribution is that **a single generative model can perform camera-controlled video generation, novel view synthesis, and camera pose estimation within a unified framework**. We acknowledge that VGGT has capabilities our model does not, such as generating 3D pointmaps.
>
> The motivation for joint learning becomes clear when considering practical issues of using off-the-shelf pose estimators as input to generation:
>
> **(1) Scale misalignment.** External pose estimators such as VGGT or COLMAP produce camera parameters at an arbitrary scale. Many existing approaches normalize cameras based on 3D point clouds, but in real-world applications we may not have access to 3D point clouds directly.
>
> **(2) Error accumulation.** Errors in input camera poses propagate and compound during generation. A pipeline of separate models has no mechanism to self-correct.
>
> Our unified model sidesteps both issues: pose estimation and generation share the same learned scale and are trained jointly. We demonstrate this through our self-consistency test [https://raysaspixels2026-icml.github.io/#table1], our model predicts camera poses from videos, then generates novel views conditioned on its own predicted poses, achieving strong results across all metrics. This closed-loop consistency is something a pipeline of separate models cannot guarantee.
>
> ### W2: Suspicious baselines in Table 3: high FID/FVD but similar or better Rerr/Terr on DL3DV-140
>
> The explanation lies in how these metrics capture different aspects of generation quality:
>
> **Rotation /  Translation Error** measures whether the generated trajectory matches ground-truth. A model can achieve low trajectory error while producing visually poor frames, for example, blurry content that nonetheless moves in the right direction.
>
> **FID/FVD** measures visual quality and temporal coherence. Our model prioritizes visually plausible content, which sometimes requires hallucinating in occluded regions. These hallucinated pixels, though visually coherent, can cause the estimated trajectory to diverge from ground-truth.
>
> In other words: low Rotation Errors and Translation Errors with high FID/FVD means the camera moved correctly but the video looks bad, while our results show higher visual quality with a trade-off in trajectory adherence. We have added visual comparisons with Kaleido on DL3DV-140 at [https://raysaspixels2026-icml.github.io/#videos-kaleido]. We encourage the reviewer to examine the qualitative differences.
>
> ### W3a & W3b: Why raxel over Plücker? Why DSCA? Why not concatenate along feature dimension?
>
> We have conducted controlled ablations for both components, keeping the model architecture fixed and swapping one component at a time:
>
> | Method | FID ↓ | FVD ↓ | Rerr ↓ | Terr ↓ |
> |---|---|---|---|---|
> | Ours (Full Model) | 7.33 | 68.17 | 0.020 | 0.018 |
> | w/o DSCA | 8.69 | 77.08 | 0.048 | 0.052 |
> | Plücker Embedding | 21.97 | 333.56 | 0.241 | 0.430 |
>
> **Raxel vs. Plücker.** Plücker embeddings have 6 channel dimensions and cannot be encoded through the spatio-temporal VAE, instead, we project them to the same dimensions as video latent tokens and concatenate them token-wise. Raxels naturally live in a shared latent space with video tokens: 3 channels pass through the same spatio-temporal VAE as RGB frames, enabling joint denoising of camera and video information. Plücker embeddings bypass the VAE and prevent this tight coupling. The ablation confirms that this shared latent space is critical.
>
> **Token vs. feature dimension concatenation.** Token concatenation provides the flexibility of doing novel view synthesis or predicting camera poses, enabling multi-task capability. Feature dimension concatenation is not as flexible, the token cannot have two different diffusion steps.
>
> **Why DSCA over standard attention.** Removing DSCA degrades all metrics. The modality collapse we originally described was observed under normalized (non-metric-scale) training. Under metric-scale training, collapse is mitigated but DSCA still contributes meaningfully, providing a useful inductive bias for modality separation. We will clarify this distinction in the revised paper.
>
> Full ablation table and qualitative comparisons: [https://raysaspixels2026-icml.github.io/#videos-ablation] (see also our response to Reviewer 26cW).

---

> > ### Author Rebuttal · Reviewer_zWsM · 2026-04-04
> >
> > Thanks for response. My concerns are largely solved. One thing I would suggest is to weaken the contribution of Raxel since 3 channels representation is better than plucker only under specific constraint, which might not transfer to other settings.
> >
> > I would like to raise my score to 4 therefore.

---

### Official Review · Reviewer_7Hcj · 2026-03-10

**Soundness:** 2
**Presentation:** 3
**Significance:** 3
**Originality:** 3
**Overall Recommendation:** 4
**Confidence:** 4

**Summary:**

To bridge the two related tasks of predicting camera trajectories and rendering novel views , this paper proposes "Rays as Pixels," a specialized video diffusion model that learns a joint distribution of videos and camera trajectories. This unified framework successfully enables joint video and pose generation, pose-conditioned video generation, and camera pose estimation. Specifically, the authors introduce the raxel representation to encode camera parameters as dense ray pixels aligned in the RGB space. Furthermore, the paper employs a decoupled self-cross attention mechanism to facilitate effective interaction between the visual and geometric modalities while preventing mutual interference.

**Compliance With Llm Reviewing Policy:**

Affirmed.

**Final Justification:**

The authors have addressed my concerns, so I maintain my positive score.

**Key Questions For Authors:**

I currently lean towards a Weak Accept due to the novel raxel formulation and strong results. However, the lack of solid ablations makes it difficult to verify if the performance gains stem from the proposed modules or from implementation tricks.

To improve the soundness of the paper, please address the following in the rebuttal:
1. Raxel Ablation: Provide a controlled ablation study isolating the raxel representation (i.e., keeping the overall network architecture fixed while only swapping the camera trajectory encoding).
2. Sparse Target Latent Ablation: Provide an ablation or deeper justification for explicitly hardcoding sparse target latents ($z_t$) into the architecture.
3. Terminology: Clarify and unify the terminology regarding the TAE/spatial encoder $\mathcal{E}$.

**Limitations:**

yes

**Strengths And Weaknesses:**

Strengths:
1. Innovative Camera Representation: The proposed raxel effectively resolves the channel mismatch of Plücker Embeddings and the depth reliance of Pointmaps. The accompanying Orthogonal Procrustes-based algorithm for parameter recovery is an elegant, closed-loop solution.
2. Strong Empirical Results: The method achieves state-of-the-art quantitative performance and demonstrates high-quality qualitative visualizations for camera-controlled video generation.

Weaknesses:
1. Under-explained Sparse Target Latents: The necessity of explicitly incorporating sparse target latents ($z_t$) is poorly explained. Why not simply employ alternating training on sparse and dense data? Ablations are needed to justify this design.
2. Lack of  Ablations for Raxels: Comparing against entirely different baselines violates the single-variable control principle. A strict ablation keeping the model architecture fixed while swapping only the trajectory encoding is crucial to prove effectiveness.
3. Inconsistent Terminology: The term "TAE Encoder" is unconventional and requires clarification. Furthermore, the encoder $\mathcal{E}$ is inconsistently described as both a "Temporal Autoencoder" and a "spatial encoder".

---

> ### Author Rebuttal · Authors · 2026-03-30
>
> We thank the reviewer for the positive assessment of our raxel formulation, the Orthogonal Procrustes recovery algorithm, and our empirical results. We address each concern and key question below.
>
> ---
>
> ### W1 / Q2: Sparse Target Latents
>
> We thank the reviewer for raising this point. The sparse target latent design is a deliberate architectural choice driven by three considerations:
>
> **(1) Fixed token count for consistent attention.** We maintain a fixed number of tokens throughout training by setting the maximum number of sparse input latents to 5. When fewer source images are provided (1–4), the remaining slots are filled with sparse target latents. This ensures the attention matrices have consistent dimensions across all training samples, which stabilizes training.
>
> **(2) Compatibility with the video diffusion backbone.** Our model is built on a video diffusion model (Wan2.1) that operates on temporally continuous frame sequences. Dense video generation is the natural output format of this backbone. The raxel representation is also inherently more informative in the dense setting and easier to learn during training, continuous camera trajectories encoded as raxels provide smooth, learnable signals across adjacent frames, analogous to how SLAM benefits from temporal continuity to track camera motion incrementally. Training only on sparse views would be akin to Structure-from-Motion, where the model must infer geometric relationships from isolated frames without the benefit of smooth inter-frame transitions.
>
> **(3) Flexible inference with arbitrary input views.** At inference time, users may provide anywhere from 1 to 5 source images. The sparse latent mechanism provides a unified interface: source images occupy their slots, and the model generates the remaining frames. Alternating between separate sparse and dense training modes would require the model to implicitly learn two different generation paradigms, whereas our design handles both through a single, consistent architecture.
>
> This is not a case where alternating training was considered and rejected — the design follows naturally from the requirements above. We will clarify this motivation more explicitly in the revised paper.
>
> ---
>
> ### W2 / Q1: Lack of Ablations for Raxels
>
> We agree that comparing against entirely different baselines is insufficient to isolate the contribution of the raxel representation. We have now conducted exactly the controlled ablation the reviewer requested: we keep the model architecture fixed and swap the camera trajectory encoding between our raxel representation and Plücker embeddings. For Plücker Embedding, which has 6 channel dimensions, we cannot encode them using Temporal VAE. Instead, we directly embed Plücker Embedding to the same dimensions of video latent tokens and concatenate them token-wise, with the rest of the architecture held constant. Both variants were trained on the same data (RE10K + DL3DV) with metric-scale alignment.
>
> We evaluate via a self-consistency test — predict camera poses from videos, then generate novel views conditioned on the predicted poses and 3 input images:
>
> | Method | FID ↓ | FVD ↓ | Rerr ↓ | Terr ↓ |
> |---|---|---|---|---|
> | Ours (Raxel) | 7.33 | 68.17 | 0.020 | 0.018 |
> | Plücker Embedding | 21.97 | 333.56 | 0.241 | 0.430 |
>
> The raxel representation outperforms Plücker embeddings by a large margin. We attribute this to the raxel being embedded in the same latent space as the video tokens, enabling joint learning of camera and appearance, whereas Plücker embeddings, operating outside of the Temporal VAE, have limited ability to jointly denoise along with video latents.
>
> We have additionally ablated DSCA and the cosine similarity loss under the same controlled setup (see our response to Reviewer 26cW for the full table and discussion). Qualitative comparisons are available at [https://raysaspixels2026-icml.github.io/#videos-ablation].
>
> ---
>
>
> ### W3 / Q3: Inconsistent Terminology (Spatial Encoder, Temporal Encoder -> VAE Encoder)
>
> We apologize for the confusion. We use the spatio-temporal VAE from Wan2.1. In our framework, we use only its encoder pathway to map input images into the latent space, which is why we sometimes referred to it as a "spatial encoder." We will unify the terminology in the revised paper to consistently use "VAE Encoder" throughout, making clear that we use only the encoding pathway.

---

> > ### Author Rebuttal · Reviewer_7Hcj · 2026-04-03
> >
> > My concerns are resolved.

---

### Official Review · Reviewer_26cW · 2026-03-12

**Soundness:** 3
**Presentation:** 3
**Significance:** 3
**Originality:** 3
**Overall Recommendation:** 4
**Confidence:** 4

**Summary:**

The paper introduces a new method for camera-controlled video generation by encoding camera information via raxels (a newly proposed dense representation of camera parameters), which is combined within the diffusion model as extra information. The paper demonstrates that this method of training is able to perform multiple types of tasks in addition to the camera-controlled generation, such as camera pose estimation and novel view synthesis, while achieving quite high performance both in terms of consistency with the source image, as well as trajectory adherence. The method is compared with a variety of camera-controlled generation methods to demonstrate its effectiveness.

**Compliance With Llm Reviewing Policy:**

Affirmed.

**Final Justification:**

All my concerns have been addressed, and I believe the method is novel and interesting enough, plus the experimental verification is solid enough, to be worth accepting.

**Key Questions For Authors:**

I do not have any additional questions, outside of the concerns raised in the weaknesses above.

**Limitations:**

Yes

**Strengths And Weaknesses:**

Strengths:
1) The raxel idea is novel and interesting, a very unique method for encoding camera information that could be useful for many other methods. It seems to work very well for camera control, a very relevant topic in video generation currently.

2) The consistency of the videos rendered with the new camera trajectories is very good, and the qualitative results are very good.

3) The method is well-explained with sufficient detail for each component of the model.

4) The ability to use the model to predict camera poses as well is quite interesting (and is comparable with VGGT, showing quite high accuracy).

Weaknesses:
1) My main concern (based on the videos included in the supplementary) is the low amount of camera motion found within the generated videos. This could make the task easier, as the results on full datasets compared to the baselines are not as impressive (see Weakness 3 below).

2) The paper mentions an additional cosine similarity loss added to the flow matching, which is claims helps with directional alignment. However, the authors do not provide any justification for this claim (no citation to similar ideas in the literature, and no ablation to show that it actually does improve the results). Also, other parts of the model need ablating (for example, the paper claims that regular self-attention leads to modality collapse, but shows no qualitative or quantitative evidence of this).

3) The results in Table 3 seem to show that across all datasets, this method is not state-of-the-art in trajectory adherence (and in the case of DL3DV-140, is in fact one of the worst models).

---

> ### Author Rebuttal · Authors · 2026-03-30
>
> We thank the reviewer for the thoughtful evaluation and for recognizing the novelty of the raxel representation, the quality of our qualitative results, the multi-task capability of our framework, and the clarity of our method description. We address each concern below.
>
> ---
>
> ### W1: Low camera motion in supplementary videos
>
> We appreciate this observation. The supplementary videos in the original submission were selected to showcase visual quality and consistency. We have now added new results with **large viewpoint changes** on DL3DV-140, compared side-by-side with Kaleido: [https://raysaspixels2026-icml.github.io/#videos-kaleido]. These are **single-image** novel view generations which is the most challenging setting, and it clearly shows how the model behaves when target views are far away from the input view.
>
>
> We note that our model is initialized from Wan2.1, which targets 16fps. The DL3DV-140 evaluation frames are subsampled from full video sequences (with camera parameters obtained via COLMAP), meaning adjacent evaluation frames can have substantial baseline differences compared to 16fps videos. Our model covers these large camera displacements while maintaining temporal smoothness, which can give the visual impression of moderate motion despite significant viewpoint changes.
>
> ---
>
> ### W2: Cosine similarity loss justification + ablation of components
>
> We agree this was insufficiently supported in the original submission. We have now conducted a full ablation study via a **self-consistency test**: we extract camera parameters from input videos, then generate novel views conditioned on the predicted poses and 3 input frames (frames 0, 16, 32 out of 65). All variants were trained on metric-scale aligned RE10K + DL3DV, and validated on DL3DV-140 videos sampled with 12fps.
>
> | Method | FID ↓ | FVD ↓ | Rerr ↓ | Terr ↓ |
> |---|---|---|---|---|
> | Ours (Full Model) | 7.33 | 68.17 | 0.020 | 0.018 |
> | w/o DSCA | 8.69 | 77.08 | 0.048 | 0.052 |
> | w/o Cosine Similarity Loss | 9.48 | 97.84 | 0.058 | 0.094 |
> | Plücker Embedding | 21.97 | 333.56 | 0.241 | 0.430 |
>
> Removing the cosine similarity loss degrades both visual quality and trajectory adherence, confirming our claim about directional alignment. Decoupled Self-Cross Attention (DSCA) contributes meaningfully across the metrics, and the raxel representation vastly outperforms Plücker embeddings, suggesting that the raxel representation is important for convergence.
>
> The degradation in ablated variants stems from two compounding factors: 1) less accurate camera pose estimation, and 2) inconsistency between predicted poses and generated videos, which is precisely what the self-consistency evaluation captures.
>
> Regarding the modality collapse claim for standard self-attention: we found that this issue was mitigated when training with metric-scale data, suggesting DSCA compensates for information lost under normalized representations. We will clarify this nuance in the revised paper.
>
> Qualitative video comparisons are available at [https://raysaspixels2026-icml.github.io/#videos-ablation]. We will add further examples in the final version.
>
> ---
>
> ### W3: Trajectory adherence not SOTA in Table 3, especially DL3DV-140
>
> We acknowledge that our method does not achieve the best trajectory adherence on every dataset. However, we would like to highlight several points:
>
> **(1) Multi-task capability within a single model.** Unlike prior methods designed solely for camera-controlled generation, our model jointly supports camera pose estimation and camera-controlled video generation within a unified framework, all without task-specific architectures or separate modules. This breadth involves trade-offs: methods specialized for a single task can optimize entirely for trajectory adherence, while our model balances multiple objectives.
>
> **(2) DL3DV-140 is a particularly challenging benchmark** due to large viewpoint changes and diverse scenes. The dataset subsamples frames from full video sequences to run COLMAP, so adjacent evaluation frames can have large baselines that are far from temporally continuous. We have added direct comparisons with Kaleido on this dataset [https://raysaspixels2026-icml.github.io/#videos-kaleido], showing superior visual quality despite the trajectory adherence gap. Furthermore, in occluded regions, generative models must hallucinate while maintaining consistency on visible areas, and the generated pixels can cause the estimated trajectory to diverge from ground-truth even when the output is visually plausible.
>
> **(3) The self-consistency evaluation (W2 above) provides a complementary perspective.** Our model achieves strong self-consistency results when conditioned on its own predicted poses, a property that trajectory adherence against GT poses alone does not capture.
>
> We will incorporate all ablation results and additional comparisons into the revised paper. We hope these new results address the reviewer's concerns.

---

> > ### Author Rebuttal · Reviewer_26cW · 2026-04-03
> >
> > With the newly provided videos and ablation studies, my concerns have been fully addressed. I think the paper and proposed method is novel and interesting enough to support acceptance.

---

### Decision · Program_Chairs · 2026-04-30

**Decision:**

Accept (regular)

**Comment:**

All reviewers are consistently positive. The AC also read all comments and agree with reviewers.